# NUP214 in Acute Myeloid Leukemia

**DOI:** 10.3390/cells14181461

**Published:** 2025-09-18

**Authors:** Øystein Bruserud, Håkon Reikvam

**Affiliations:** 1Acute Leukemia Research Group, Department of Clinical Science, University of Bergen, 5007 Bergen, Norway; hakon.reikvam@uib.no; 2Section for Hematology, Department of Medicine, Haukeland University Hospital, 5009 Bergen, Norway

**Keywords:** nuclear pore, leukemogenesis, chemosensitivity, fusion protein, nucleoporin 214, DEK proto-oncogene, SET nuclear proto-oncogene, ABL proto-oncogene 1, non-receptor tyrosine kinase, sequestosome-1

## Abstract

Nucleoporin 214 (NUP214) is a component of the nucleopore molecular complex, but in addition to this role in nucleocytoplasmic transport it is also involved in the regulation of gene transcription/translation, intracellular signaling, cell cycle progression and programmed cell death. Several uncommon translocations associated with acute myeloid leukemia (AML) involve the *NUP214* gene, and the corresponding fusion proteins are involved in leukemic transformation. First, the t(6;9) translocation encodes the DEK-NUP214 fusion protein; this translocation is seen in 1–2% of AML patients and is associated with an adverse prognosis that is improved by allogeneic stem cell transplantation. Second, the *SET-NUP214* fusion gene is less common in AML and is formed either by del(9)(q34.11q34.13) or a balanced t(9;9)(q34;q34). This AML variant shows several biological similarities with the *DEK-NUP214* variant, but the possible prognostic impact of this fusion protein is not known. Finally, the *NUP214-ABL1* and especially the *NUP214-SQSTM1* fusions are very uncommon, and only a few case reports have been published. In this article, we review the functions of the genes/proteins formed by these fusion genes, the available studies of molecular mechanisms and biological functions for each fusion protein, the characteristics of the corresponding AML cells, the clinical characteristics of these patients and the possible prognostic impact of the fusion genes/proteins.

## 1. Introduction

NUP214 is a component of the nuclear pore molecular complex that has a key role in molecular transport across the nuclear membrane [1,2,3]. Targeting of this complex, and thereby the molecular export from the nucleus to the cytoplasm, is now regarded as a possible therapeutic strategy in human AML, and pharmacological inhibition of Exportin 1/XPO1 has been investigated in several clinical studies [4]. This therapeutic approach has antileukemic effects even when used as monotherapy [5,6], but high toxicity has been a problem in certain studies of combined treatment with conventional chemotherapy [7]. However, the therapeutic experience strongly suggest that nuclear/cytoplasmic transport is important for both leukemogenesis and chemosensitivity in human AML. This importance in leukemogenesis is also illustrated by the AML-associated genetic abnormalities involving genes/proteins that are important in nuclear export/import [8,9], including the NUP214-involving abnormalities that are described and discussed in this review [10].

## 2. Methodology

This review is based on selected publications from the PubMed database (accessed last time 17 September 2025). All articles identified by using the combined searching words acute, myeloid, leukemia, and NUP214, were screened and form the basis of the review; additional references were included to further explain the biological and clinical context of NUP214 and NUP214-containing fusion proteins identified in human AML.

## 3. The Structure and Function of the Nuclear Pore Complex and the NUP214 Molecule

### 3.1. The Structure of the Nuclear Pore Complex

The structure and function of the nuclear pore complex has been described in detail in several excellent reviews [1,2,3,11,12,13]; for this reason, we will only give a brief overview of the general structure and then focus on NUP214. The nuclear pore complex is formed by approximately 30 distinct nucleoporins that are organized into various subcomplexes. The nucleoporins are therefore classified into various groups (Figure 1) [1,2,3]:*The general structure.* The main structures is the spoke complex (also called the central scaffold/framework) that forms the central structure with a central pore that facilitates macromolecular passage between nucleus and cytoplasm. This central complex is flanked by the cytoplasmic and the nuclear rings. Filamentous structures project from both these rings; they are referred to as the cytoplasmic filaments and the nuclear basket, respectively. Based on this main structure of the nuclear pore complex, the nucleoporin proteins can be classified into various subsets as described below.*Scaffold proteins.* These proteins form the central framework or complex of the pore, and several of these proteins span both the cytoplasmic and nuclear side of the pore. Nucleoporins form the core of the complex.*The transmembrane ring nucleoporins.* These proteins are embedded within the nuclear envelope, and they thereby anchor the pore complex to the nuclear membrane.*Peripheral proteins.* The peripheral proteins are associated with the cytoplasmic filaments of the pore and with the nuclear basket on the nuclear side. These proteins function as important regulators of the selectivity/permeability of the pore.*Proteins with FG motifs.* Some authors prefer to distinguish a fourth separate subgroup among the peripheral proteins, namely nucleoporins that contain repetitive phenylalanine-glycine (FG) motifs (GLFG or FXFG motifs) interspersed between 20 charged amino acid residues. The FG domains are important both for anchoring of the pore complex to the nuclear membrane and for regulation of the transport through the pore complex. Approximately one-third of the nucleoporins contain FG domains. NUP214 is classified as a peripheral FG-containing nucleoporin.

It has been emphasized that the nuclear pore complex has a dynamic structure that differs depending on the biological status of the cell, e.g., it differs/adapts both to cell division and terminal differentiation [3].

### 3.2. The Structure and Function of NUP214

NUP214 is classified as a peripheral nucleoporin that is anchored to the cytoplasmic ring of the nuclear pore complex and contains an FG domain; it contributes to the formation of the cytoplasmic filaments of the pore complex and is thereby involved in the regulation of molecular transport [1,2,3,11,12,13]. It interacts with NUP98 as well as multiple transport receptors, and thereby, it mediates their passage through the nuclear pore complex [13]. The interaction partners include the main protein exporter XPO1 as well as RNA exporters including the main mRNA exporter NXF1. NUP214 is also classified as a peripheral nucleoporin, a group of nine molecules that are involved in the regulation of selectivity and permeability of the nuclear pore [3].

As described in Section 3.1 the nuclear pore is a large molecular complex where NUP214 is located on the cytoplasmic side [13]. The NUP214 molecule is composed of three domains (Figure 2) [14,15,16] (for detailed information and additional references see the UNIPROT database https://www.uniprot.org; accessed 13 August 2025). First, the N-terminal β-propeller domain is involved in molecular interactions within the nuclear pore complex. Second, the central coiled-coil motifs represent two central coiled-coil regions that anchor NUP214 to the nuclear pore complex through their interactions with the NUP98 molecule. Third, C-terminal FXFG domains can be detected on both sides of the nuclear pore complex [14,16]; NUP214 is located at the cytoplasmic side of the nuclear pore and its FXFG domains has several binding sites for XPO1 and also binding sites for RNA export; the interaction with XPO1 induces stabilizing conformational changes in both proteins [17]. NUP214 thereby functions as a docking site for XPO1 export complexes on the cytoplasmic side of the nuclear pore complex [17].

NUP214 depletion results in nuclear accumulation of XPO1 client proteins, including cargoes with the specialized NES (Nuclear export signal) molecular binding motif that mediates binding to XPO1 [4]. NUP214 is localized on the cytoplasmic side of the nuclear pore complex, the schematic structure of the nuclear pore complex is shown in Figure 1.

NUP214 is a part of the cytoplasmic filament nucleoporin complex together with NUP98; another nuclear pore molecule that is also involved in AML-associated genetic abnormalities [18,19]. As pointed out in previous reviews [12], several proteins of the nuclear pore complex have diverse functions, and this is true also for the NUP214 protein as well as its interaction partner in the nuclear pore complex (e.g., XPO1, NUP98):*NUP98 anchoring.* NUP98 anchors NUP214 to the nuclear pore (see above), but NUP98 can in addition modulate the expression of genes involved in regulation of cellular development and cell proliferation [20].*XPO1 colocalization.* Overexpressed NUP214 colocalize together with XPO1 and importin 1; this leads to depletion of these two molecules from the nuclear pore complex and finally growth inhibition and induction of apoptosis [21].*NUP214 knockdown.* The functional effects of NUP214 have been investigated in various cellular models. Knockdown of NUP214 leads to cell cycle arrest in G_2_ phase in embryonic cells [22], whereas high levels/overexpression causes G_0_ phase arrest, nucleocytoplasmic transport defects, XPO1 depletion from the nuclear pore complex and apoptotic cell death [21]. These observations further illustrate the complexity of the NUP214 effects on fundamental cellular processes, but it is not known whether (any of) these effects are relevant in human AML.*Scaffold for signaling proteins.* Nucleoporins often function as scaffolds for proteins in intracellular signaling; and this is true also for NUP214. Transforming growth factor β (TGFβ) signaling can be mediated by an interaction between SMAD2 and NUP214, the same is true for SMAD3 and SMAD4 that also bind to NUP214 [23,24,25,26]. TGFβ can be involved in human leukemogenesis [27,28]. Furthermore, NUP98 is a binding partner both for NUP214 and RAE1, and this molecular complex can be recruited to the kinetochore as well as the spindle and spindle poles during mitosis [11,12].*Noncoding RNA regulation.* NUP214 expression is also regulated by noncoding RNAs; NUP214 expression can be downregulated by miR-133, and this mechanism may contribute to carcinogenesis [29].

To conclude, NUP214 seems to have multiple and complex cellular effects. Deregulation of NUP214 affects the nuclear transport, but the main architectural structure of the nuclear pore complex is not affected [13,21,22,29]. It is in addition involved in regulation of intracellular signaling, cell cycle progression/mitosis and programmed cell death even in cancer cells; its fundamental cellular importance is further illustrated by the observation that genomic knockout of *NUP214* leads to embryonic lethality in mice [21,22].

## 4. Activation of HOX Genes in AML: A Common Characteristic of Several AML Subsets That Can Be Mediated by Nuclear Pore Molecules

The HOX proteins are DNA binding proteins that regulate gene expression programs that are important in hematopoiesis [30,31]. The nuclear pore molecule XPO1 is involved in the regulation of HOX gene expression through recruitment of AML-associated nucleoporin fusion proteins as well as mutated NPM1c to chromatin-bound XPO1 [32,33,34].

The ParaHox gene CDX2 increases the expression of several HOX genes [35]. The CDX2 expression is regulated by a wide range of intracellular mediators [36], including the potential tumor suppressor miR-196b [37], and CDX2 expression has been identified as the transforming event in a murine AML model [38]. However, human AML is often characterized by low miR-196b expression [37] but simultaneously high leukemogenic HOX gene expression [35,37,39]. This altered balance between the two effects is consistent with the hypothesis that the effect of miR-196b on CDX2 expression is less important in most cases of human AML [35]. Detectable AML cell expression of CDX2 is seen for a large majority of AML patients [35]; the levels are particularly high for patients with normal karyotype whereas the favorable abnormality t(8;21) shows lower levels [39].

Experimental studies suggest that the CDX2-related mediator CDX4 has minimal effects on normal hematopoiesis in adults [40]. However, CDX4 seems to participate in MLL-AF9-mediated leukemogenesis because it shortens disease latency, although it is not essential for development of this leukemic disease and does not alter the expression of HOX genes in this AML model [40]. A leukemogenic contribution/effect was also suggested by another study that described that overexpression of CDX4 in murine progenitors generated a partially penetrant disease with long latency, but the HOX cofactor MEIS1A accelerated leukemogenesis and made the disease fully penetrant [41]. However, CDX4 as well as CDX1 are detectable in AML cells only for a minority of patients (~25%) and normal karyotype AML shows low/undetectable CDX1/CDX4 levels [39,41]. Thus, CDX2 seems important for regulation of HOX gene expression in human AML cells, whereas CDX4 and possibly also CDX1 seem less important or only contribute to leukemogenesis in certain AML subsets. Finally, it should be emphasized that these CDX molecules also regulate the expression of several non-HOX genes (including transcriptional regulators) [42,43]. Finally, HOX gene expression is also regulated by other mechanisms than CDX molecules, including Glycogen synthetase kinase 3 (GSK3)/CREB/MEIS1 [44] and Disruptor of telomeric-silencing 1-like (DOTIL) that catalyzes histone methylation and seems important for maintenance of HOXA gene expression in *MLL*-mutated/translocated AML [45].

To conclude, HOX gene expression seem to be important for leukemic transformation in various subsets of AML, including both DEK-NUP214 (Section 5) and SET-NUP214 (Section 6) AML [32,33,34].

## 5. The t(6;9)(p22;q34) Translocation and the DEK-NUP214 Fusion Protein

The t(6;9)(p23;q34) translocation is uncommon in AML; it is detected in only ~1% of adult [46] and approximately 2% of pediatric patients (for references see [47]). This translocation forms the DEK/NUP214 fusion gene/protein.

### 5.1. The Structure and Function of the DEK Proto-Oncogene Protein

#### 5.1.1. The Structure and Function of the DEK Proto-Oncogene Protein

The structure and function of DEK has been described in several recent reviews [47,48,49,50,51,52,53,54,55,56]; it is a DNA binding protein that is also important in the architectural control of chromatin (Table 1, Figure 3). In brief, the gene is located on chromosome 6p22–23 and encodes a 375-amino acid protein (43 kDa) with four distinct stretches of acidic amino acids. The DEK protein is able to multimerize; the amino acids 270–350 seem to be particularly important for the multimerization that also depends on phosphorylation by the CK2 kinase [53]. Two molecular regions seem to be responsible for DNA binding. One region is the SAF-box DNA binding motif between amino acids 149 and 187, and the other region overlaps the multimerization domain and is located between amino acids 270 and 350. These two DNA binding regions differ with regard to their binding properties and the effect of CK2-mediated phosphorylation.

DEK is a nuclear protein; the protein levels are high in proliferating cells but very low in resting and terminally differentiated cells [55]. The cellular levels do not change much during cell cycle progression, but the phosphorylation status shows a considerable variation during cell cycle progression with the highest levels in the G1 phase.

#### 5.1.2. Cellular DEK Is a Histone Chaperone

DEK should also be regarded as a histone chaperone, i.e., it can bind to the histone octamer surface, as it is located at the nucleosomes [47,56]. The DEK interaction with the octamer is mediated by the DEK amino acid residues 52–68, a molecular region referred to as the histone-binding region [56]. There seems to be a preferential binding of DEK to transcriptional start sites where it colocalizes with H3 trimethylation and stimulates methyltransferase activity [56]. DEK may thereby suppress transcription. However, DEK is also important in the regulation of histone H3 and to a lesser extent histone H4 acetylation through its interactions with both histone deacetylases and acetyltransferases [58,59,60]. Finally, DEK is in addition important in DNA repair [61,62] and for RNA splicing [63,64,65,66].

#### 5.1.3. Other Important Functions of the Cellular DEK Oncoprotein

The overall effects of the DEK protein in various cell types are presented in more detail in Table 2 [53,54,55,56,57,58,59,60,61,62,63,64,65,66,67,68,69,70,71,72,73,74,75,76,77,78,79,80,81,82,83]; based on these observations, the following main conclusions can be made:*DNA/RNA binding.* DEK can bind both DNA, RNA and chromatin proteins [54,55,64]; its binding affinity to both DNA and histones/chromatin is regulated by post-translational phosphorylation and acetylation [54,67]. It functions as a regulator of both histone acetyltransferases and deacetylases [58,59,60].*Signaling target molecules and functional cellular effects.* Among its targets/molecular interaction partners are RAD51 [62], STAT5 [73,74], mTOR [58], IRAK1 [69], p300 [79], p53 [73,74] and NCOR1 [58]. DEK is thereby involved in the regulation of several fundamental molecular processes including DNA repair [61,62], epigenetic regulation/histone acetylation [58,59,60], RNA processing and splicing [63,64,65,66], and p53 stabilization [73,74]. DEK becomes important for the cellular regulation of proliferation/cell cycle progression [64,69,70,71], survival/apoptosis (it has an antiapoptotic effect) [64,69,70,71,73], cellular quiescence [47,58,71,72], expression of several oncogenes and tumor suppressor genes [64], and cellular adhesion/migration [69,70,71].*Extracellular release.* Extracellular release of DEK can be a part of the communication between neighboring cells (see Section 5.2) [81,82,83].*Leukemogenesis.* DEK can possibly also regulate/support AML leukemogenesis both through direct effects in the leukemic cells [78] and indirect effects on supporting neighboring non-leukemic cells (e.g., M2 macrophages, endothelial cells/angiogenesis) [70,79].

### 5.2. Extracellular Release of a Soluble Form of DEK

Posttranscriptional modulation of DEK including phosphorylation (clustered at the C-terminal region) and poly(ADP-ribosyl)ation removes DEK away from DNA/chromatin. However, to the best of our knowledge it is not known whether/how secreted DEK shows posttranscriptional modification. It is secreted (i) passively during apoptosis or (ii) actively by macrophages after IL1 stimulation in a Golgi-independent way as either a free form or within exosomes [54,82]. This extracellular form can function as a chemoattractant for both neutrophils, T cells and NK cells. The extracellular form can also be endocytosed after binding to heparan-sulfate-like proteoglycans [83], it can be detected in serum especially in patients with inflammatory diseases [84,85], and it seems to function as a regulator of hematopoiesis (for details see [52,86]). Effects on hematopoiesis seem to be mediated by ligation of the CXCR2 chemokine receptor [49,87].

Extracellular DEK can also be components of neutrophil traps [80,81]. Animal models suggest that this soluble form is important in autoimmune diseases, and its proinflammatory activity can then be modulated by DEK-targeting DNA aptamers [80].

It should finally be emphasized that it is not known whether soluble DEK has a role in leukemic hematopoiesis, for example, a direct effect on the AML cells or in the regulation of the AML-supporting effect by certain immunocompetent cells. Although AML cell expression of the CXCR2 receptor is associated with an adverse prognosis in human AML [88,89,90,91], it is not known whether soluble DEK contributes to this together with the several CXCL chemokines that also bind to the receptor.

### 5.3. Expression of Normal DEK in AML

Normal DEK is expressed by primary AML cells, but the mRNA level in AML cells is generally lower than in normal bone marrow cells [78,92]. One of these studies also suggests that the DEK protein level differs between AML subsets; patients with favorable prognosis showed particularly low levels whereas the highest levels were detected for patients with an intermediate phenotype [78]. However, other studies have shown conflicting results with even DEK mRNA overexpression for a large majority [92] or for a subset [93] of AML patients. The mRNA expression of DEK seemed to be associated with differentiation and CD34^-^ AML cells showing higher levels than CD34^+^ cells [92].

### 5.4. The Structure of the DEK-NUP214 Fusion Protein

The full-length DEK protein consists of 375 amino acids and has several acidic regions, neighboring SAP and pseudo-SAP domains, an RNA binding region including a specialized subregion called the nuclear localization region, and an unique C-terminal DEK domain responsible for DNA binding and multimerization to chromatin (for references and details, see Figure 3 and Section 5.1 [13,47]). The DEK-NUP214 fusion protein includes amino acids 1–349 of the DEK protein—only the last C-terminal amino acids 350–375 are missing; these amino acids are fused to the aminoterminal amino acids 813–2090 of the NUP214 molecule (Figure 3). The fusion protein thereby includes the RNA binding/nuclear localization signal region as well as the chromatin/multimerization region of DEK together with the C-terminus FG repeat region of NUP214. This represents a main structure similarity to the SET-NUP214 fusion protein, i.e., a large fragment from the C-terminal part of the NUP214 fused to a nearly complete chromatin-remodeling protein [13,47].

### 5.5. Biological Mechanisms Involved in the Leukemic Transformation of AML Cells with t(6;9)

Relatively few studies have investigated the molecular mechanisms behind the leukemic transformation in cells with the t(6;9) translocations. The most important observations are summarized in Table 3 [47,94,95,96,97,98,99,100,101,102,103,104,105,106] and will also be stated below. Both the DEK-and NUP214-derived parts of the fusion protein are responsible for these effects.

#### 5.5.1. Effects on the Level of and the Interaction with the Full-Length DEK Protein

The localization of full-length DEK to histone H3 cannot be detected in the presence of the fusion protein [94,95]. Thus, there seems to be a functional interaction between full-length DEK and the fusion protein where the DEK-NUP214 protein modulates at least certain functions of full-length DEK. The fusion protein and the DEK full-length protein also show an overlap with regard to interacting/targeted proteins [47,95]. These observations suggest that there may be competition between these two proteins with regard to the binding to and possibly also their effects on common interacting targets. Variation in DEK protein levels between individual patients [78,92,93] may therefore lead to variations in the competition for common binding sites and thereby differences in the biological effects of the fusion protein between individual t(6;9) positive patients. Finally, the fusion protein causes a transformation of immature hematopoietic cells [98], whereas the full-length DEK also seems to be important for preservation of hematopoietic stem cells [58].

#### 5.5.2. Effects of the Fusion Protein on Intracellular Mediators; Effects on mTOR, NFκB, STAT5 and p53

The DEK-NUP214 protein modulates intracellular signaling, and both DEK and DEK-NUP214 target mTOR activation, NFκB signaling, STAT5 activation and the p53 status [97,99,100].

The t(6;9) patients show the activation of STAT5 [107]. Constitutive activation of FLT3 and STAT5 enhances the self-renewal and alters differentiation of hematopoietic stem cells, and experimental studies show that both these two events can be involved in leukemic transformation of hematopoietic stem cells [100]. Other studies have later confirmed that the t(6;9) fusion protein can induce a leukemic phenotype from a minor subset of hematopoietic stem cells; these transformed cells are characterized by a surface marker pattern similar to long-term hematopoietic stem cells together with STAT5 as well as STAT3 activation [101,107]. Studies in other experimental models have also shown that STAT5 is important for AML stem cell renewal [108]. Furthermore, AML cells often have *FLT3-ITD*, *FLT3*-tyrine kinase and *SCF* receptor mutations that initiate constitutive downstream signaling with STAT5 activation; most patients with t(6;9) AML show additional *FLT3-ITD*s that possibly also contribute to the STAT5 activation observed in these AML cells [109]. The *FLT3-ITD*-induced STAT5 activation in such AML cells can then be further potentiated by the FYN non-receptor tyrosine kinase [101]. STAT5 activation is also involved in the regulation of p53 degradation in AML cells [74]. Finally, STAT5 inhibition is now regarded as a possible strategy in human AML; experimental studies have shown that STAT5 inhibition increase the cytotoxic effect of JAK1/2 inhibition and combined treatment may therefore be a possibility [110]. Taken together, these observations suggest that STAT5 is important for regulation of leukemogenesis and chemosensitivity in t(6;9) AML, especially for the majority of patients with additional *FLT3* mutations.

#### 5.5.3. Effects by the NUP214 Derived Part of the Fusion Protein: Effects of Fusion Protein Binding to the Nuclear Transporter XPO1

*The involvement of XPO1.* A recent study further elucidated the dependency of DEK-NUP214 AML on XPO1 in various human AML models [10]. First, deletion of XPO1 in the DEK-NUP214 positive FKH-1 AML cell line as well as selective XPO1 inhibition by eltanexor in primary human AML cells reduced the XPO1 expression, disrupted XPO1 chromatin co-localization with the fusion protein, and induced apoptosis and cell cycle arrest. Second, the loss of chromatin binding resulted in downregulation of the target genes of the fusion protein, including proteins involved in regulation of cell cycle progression and self-renewal. Eltanexor treatment in a patient-derived *DEK-NUP214* AML xenograft model disrupted leukemia development with molecular clearance of the bone marrow after a median of 377 days, whereas control mice succumbed after a median of 244 days. XPO1 inhibition is now regarded as a possible therapeutic strategy in AML [4], and the present observations suggest that XPO1 inhibition will be particularly effective in t(6;9) AML.

#### 5.5.4. The Multiple Molecular Effects of DEK-NUP214 Influence Multiple Cellular Functions Including Transcription/Translation, DNA Repair, Metabolism and Proliferation/Survival

The fusion protein alters the regulation of gene transcription, and the mechanisms behind this effect are at least partly dependent on NUP214-mediated binding of the fusion protein to XPO1 with increased expression of several *HOXA* and *HOXB* genes [102,103,104]. HOX cluster genes are important mediators in leukemogenesis for many AML variants, and their expression is regulated by complex interacting mechanisms involving transcription factors, epigenetic regulators, noncoding RNAs and chromatin modulators [31]. The overall HOX gene expression pattern in DEK-NUP214 AML differs from other AML variants [92,104].

The fusion protein causes a global increase in protein synthesis; this seems to be an effect restricted to cells of the myeloid lineage, and it correlates with the phosphorylation status of the translation initiating protein EIF4E [99,106].

The fusion protein also binds several proteins involved in RNA processing/splicing, epigenetic regulation/chromatin remodeling, DNA repair, programmed cell death and regulation of (cytokine-dependent, mTOR mediated) proliferation/cell cycle progression [93,94,95,98,99]. These effects are partly mediated by binding to DEK interacting proteins [47,95]; regulation of these fundamental cellular processes is thus common for full-length DEK and the DEK-NUP214 fusion protein. Finally, the fusion protein causes a shift in energy metabolism [102], such effects have not been described for full-length DEK.

### 5.6. Clinical and Biological Characteristics of DEK-NUP214 AML

Relatively few studies are available with regard to the clinical and biological characteristics of patients with the t(6;9) translocation, and several of these reports include few patients [46,111,112,113,114,115,116,117]. The following observations have been made:*t(6;9) in different myeloid malignancies.* The translocation has been described both in patients with AML, MDS and blast phase of CML [46,112].*Age.* The *DEK-NUP214* variant of AML can be detected in both pediatric (31 out 59 patients) and adult (28/59) patients, but the adult patients seem to be relatively young (median age 35 years, range 17–66 years) [115,116]. Similar variations have been observed in other studies [112,113,114].*Leukemization.* Peripheral blood leukemization is relatively common (median level of AML blasts in peripheral blood 28%), but the total peripheral blood white blood cell count shows a considerable variation (median 12.9 × 10^9^/L, range 0.5–181 × 10^9^/L) between patients [46]. Wide variations have been observed also in other studies [112,113,114].*Bone marrow failure.* Anemia (median levels 8.4 g/100 mL) and thrombocytopenia (median level 45 × 10^9^/L) are common [46]. Similar cytopenias have been observed in other studies [113,114]. Furthermore, the median level of bone marrow blasts was 58% (range 4–99%) in one study [46]. Similar variations have also been observed in other studies [112,114].*AML cell morphology.* Up to half of the patients have AML cells with M2 phenotype according to the FAB classification, but the M4/M5 phenotype is also common [46,114,115,117]. Basophilia defined as ≥2% basophils in the bone marrow can be observed for a subset of patients, the same is true for Auer rods and dysplasia of single or multiple lineages [46]. A small study including 16 adult patients described trilineage dysplasia for 81% of patients [112]. Similar results have been described in other studies [46,113,114]. Finally, one previous study also described a patient with a morphological phenotype resembling acute promyelocytic leukemia and paraneoplastic manifestations consistent with Still’s syndrome [118].*Molecular signs of differentiation.* The AML blasts express CD9, CD13, CD33 and HLA-DR for almost all patients, expression of CD38, CD45 and CD117 is also common [46,112,113,114]. CD15 and CD34 are expressed for a majority of patients [46,112,113]. The immunophenotype is suggestive but not diagnostic for this AML variant, and for the evaluation of minimal residual disease (MRD), genetic PCR diagnosis therefore has to be used [111].*Other genetic abnormalities.* The t(6;9) translocation seems to be the sole cytogenetic abnormality for the majority of patients [46,112,113,114]. Complex karyotypes are uncommon [112,113]. A majority of both pediatric and adult patients seem to have *FLT3-ITD*, but genetic abnormalities in the *FLT3* tyrosine kinase domain are uncommon/absent [46].

### 5.7. The Possible Prognostic Impact of the DEK-NUP214 Fusion for AML Patients Receiving Intensive Antileukemic Treatment: Adverse Prognosis After Intensive Antileukemic Chemotherapy

One large study has investigated the effect of intensive conventional chemotherapy in t(6;9) patients [46]. This study included 69 patients previously included in 20 protocols during the time period 1987–2002. The patients constituted 0.9% of the 7690 patients included in these studies (31 children/pediatric patients, 38 adults). There was a male predominance among the 38 adult patients (71%). The t(6;9) translocation was the only cytogenetic abnormality for 61 patients, but 71% had *FLT3-ITD*. The presenting white blood cell count (median 28 × 10^9^/L, range 0–95 × 10^9^/L) was lower than the counts in the other 7671 patients. Six percent of the patients had previous MDS, 6% had secondary AML, 6% had relapsed/refractory AML and 11 adult patients later received an allotransplantation.

Outcome analyses were possible for 31 pediatric patients and 31 adult patients with previously untreated disease. The following observations were made:*Remission induction.* Complete hematological remission was achieved for 71% of pediatric patients and 58% of the adults; these rates compared similar to the corresponding rates for young adults with intermediate (76%) and unfavorable prognosis (55%) in the SWOG/ECOG S9034/3489 study [119].*Survival analyses.* The 5-year survival estimate was 28% for pediatric and 9% for adult patients. The overall survival was comparable to the overall survival for young adults with unfavorable (12%) disease in the SWOG/ECOG S9034/3489 study [119]. Furthermore, overall and AML-free survival did not differ between pediatric and adult patients.*Independent prognostic factors.* Multivariate analyses showed that the white blood cell count remained significant for decreased overall survival and high bone marrow blast count for decreased AML-free survival [119].

An adverse prognosis is also suggested by another study of adult patients [120]. Furthermore, a pediatric study included 48 t(6;9) patients treated during the time period 1988–2010; these patients were previously included in various clinical protocols [117]. The study showed a significantly lower remission rate (67% versus 79%, *p* = 0.04), increased relapse rate and poor overall survival (39% versus 57%, *p* = 0.03) for the 48 t(6;9) patients compared with other AML patients. Additional *FLT3-ITD* did not seem to have any prognostic impact [117]. Thus, t(6;9) AML is generally associated with an adverse prognosis independent of age (i.e., similar for adult and pediatric patients).

Taken together, these observations suggest that the t(6;9) abnormality is associated with an adverse prognosis and increased risk of chemoresistant AML relapse. Three other studies have also described the clinical characteristics, including therapeutic responses, in patients with t(6;9) AML [112,113,114]. The results from these studies have to be interpreted with care because the study included only 8–13 patients, but none of these studies contradict the conclusion that t(6;9) is associated with an adverse prognosis in human AML.

### 5.8. The Possible Prognostic Impact of the DEK-NUP214 Fusion for AML Patients Receiving Intensive Antileukemic Treatment: Improved Prognosis After Allogeneic Stem Cell Transplantation

Two studies based on the databases of the Japan Society of Hematopoietic Cell Transplantation and the Japan Cord Blood Bank Network have investigated the effect of allotransplantation for patients with this cytogenetic abnormality [115,116].

The first Japanese study included 64 de novo t(6;9)(p23:q34) AML patients that received their first transplant between January 1996 and December 2007 [115]. The study included 9 patients below and 57 patients above 15 years of age; the median age of the adult patients was 35 years (range 17–58 years) and 32 of them received bone marrow grafts. The 57 adult patients were analyzed in more detail. The univariate analysis of these adult patients showed that disease status (i.e., pretransplant complete hematological remission) was the only significant prognostic factor with regard to overall survival; the 3-year overall survival was then 69% versus 29% (*p* < 0.003). The cumulative incidence of relapse was also significantly lower for the complete remission patients (58% versus 25%, *p* = 0.005), whereas the non-relapse mortality did not differ between these two groups. Furthermore, disease status remained significant after multivariate analysis (*p* < 0.02), but a M2 phenotype according to the FAB classification also reached statistical significance in this multivariate analysis and was associated with a favorable prognosis (*p* < 0.003). Patients not in remission and without an M2 phenotype showed a 2-year overall survival of only 10%, but it should be emphasized that this group included only 10 patients.

The second Japanese study was a matched pair study including 59 adult patients (i.e., all >15 years of age, only two patients >55 years of age) with the t(6;9)(p23:q34) translocation; these patients were compared with 171 normal karyotype patients matched for age, disease status at transplantation and graft source [116]. However, the two groups differed with regard to conditioning with total body irradiation. Neither the 5-year overall survival (45% versus 40%), 5-year AML-free survival (42% versus 33%), 5-year cumulative incidence of relapse (42% versus 45%), nor non-relapse mortality (16% versus 22%) differed significantly when comparing t(6;9) versus normal karyotype patients.

A more recent third study investigated 195 t(6;9)(p23:q34) patients from the European Society for Blood and Marrow Transplantation (EBMT) registry; these patients were transplanted at 97 centers during the period 2006–2016 [121]:*Relapse and survival.* Patients transplanted in first complete remission had lower relapse risk as well as higher AML-free and overall survival than patients transplanted with more advanced disease (i.e., second or later remission, active residual disease). Furthermore, for patients transplanted in first remission the relapse incidence was 19%, two-year AML-free survival 57% and two-year overall survival 61%.*Prognostic classification.* The results for patients transplanted in first remission are comparable to patients with favorable to intermediate risk disease [122], whereas other patients classified as having high-risk disease show two-year relapse rates up to 60% after allogeneic stem cell transplantation (for references and more detailed comments, see [123]).*FLT3-ITD.* The presence of FLT3-ITD did not significantly influence the outcome in this study, but it should be emphasized that information about FLT3 status was only available for approximately one-third of the patients.

A smaller fourth study including 21 patients (16 allotransplanted patients) also described significantly better survival for allotransplanted patients that for patients not receiving allotransplantation with regard to overall survival [124]. These authors also suggested that FLT3 inhibition could potentially improve survival for t(6;9) AML patients, but the study was too small to support this hypothesis.

Taken together, these studies strongly suggest that the t(6;9) abnormality is associated with a better prognosis when patients can be allotransplanted, especially when transplanted in first complete hematological remission. This is also in accordance with recent clinical recommendations/guidelines [123,125,126]. Three other clinical studies have also described the clinical characteristics of t(6;9) patients, including their therapeutic responses [112,113,114]. The results from these studies have to be interpreted with great care because each of them included only 8–13 patients, but none of these studies contradict the conclusion that the prognosis is improved for AML patients with t(6;9) when allogeneic stem cell transplantation is possible.

### 5.9. The Possible Prognostic Impact of the DEK-NUP214 Fusion Pediatric AML Patients Receiving Intensive Antileukemic Adverse Prognosis Improved by Allogeneic Stem Cell Transplantation

A previous study described the clinical outcomes for 62 pediatric patients with t(6;9) AML, 54 patients had AML and 8 patients MDS [127]. Median age at diagnosis was 10.4 years, and there was a male preponderance. The outcome analyses showed 5-year event-free survival of 32%, 5-year overall survival of 53%, and 5-year cumulative incidence of relapse at 57%. Allogeneic stem cell transplantation in first complete remission (18 of the patients) improved the 5-year event-free survival (68% versus 18%, *p* < 0.01); the 5-year cumulative incidence of relapse was also decreased (13% versus 81%, *p* < 0.01), whereas the difference in overall survival did not reach statistical significance (68% versus 54%). Although the comparisons of patient subsets are based on analyses of relatively small patient groups, it seems justified to conclude that t(6;9) AML is a high-risk disease for pediatric AML patients, but the prognosis for pediatric patients seems to be improved by allogeneic stem cell transplantation.

### 5.10. Should MDS Patients with t(6;9) Rather Be Classified as AML Patients?

The t(6;9) abnormality can also be detected in MDS and not only in AML [103,128,129,130]. However, the largest study comparing 33 MDS and 74 AML patients showed that MDS patients were generally older, had lower peripheral blood blast and total white blood cell counts, higher platelet counts, and lower frequency of *FLT3* mutations. Furthermore, for patients not receiving allotransplantation, the survival was significantly longer for MDS than for AML patients (median survival 26 versus 13 months, *p* = 0.0035) but with no long-term survivors in any of the two groups. In contrast, the survival did not differ between AML and MDS patients (i.e., approximately 50% long-term survivors) that received an allotransplant at any course of their disease (64% of the patients), and the multivariate analysis including all patients showed that stem cell transplantation and initial diagnosis (AML versus MDS) were the only significant prognostic factors for survival [103]. Finally, this study also suggested that the prognosis of t(6;9) patients after allotransplantation was similar to patients with intermediate cytogenetic risk [103]. Thus, AML and MDS patients with t(6;9) show clinical/therapeutic differences [103,129] that justify the present classification in two separate entities at least until further notice.

## 6. The del(9)(q34.11q34.13) Abnormality with the SET (SET Nuclear Proto-Oncogene)-NUP214 Fusion Protein

A recent systematic review described the occurrence of the *SET-NUP214* fusion gene in patients with hematological malignancies [131]. Their analysis was based on data extraction from 35 selected articles and included patients with any type of adult hematological malignancy that expressed this fusion gene and with available complete characterization with information about treatment and outcome. This review showed that the *SET-NUP214* translocation was most often reported for T-ALL (30 patients identified in 8 selected articles), whereas only six patients were identified with either AML (3 patients), myeloid sarcoma (1 patient), acute unclassified leukemia (2 patients) and mixed phenotype acute leukemia (1 patient). Similar observations were made in a more recent systematic review including 81 patients [132], and a large clinical study described that the *SET-NUP214* abnormality occurred especially in patients with the T cell receptor (TCR) γδ variant of T-ALL [133]. A recent study also described the occurrence of *SET-NUP214* fusion in two patients with blast phase of chronic myeloid leukemia (CML) [134]. The fusion has only been described in a few published cases of B-ALL [131,132], but the T-ALL cells may express aberrant B cell [135] as well as myeloid markers [131,132]. To the best of our knowledge, this fusion gene has not been detected in patients with polycythemia vera, essential thrombocytosis or chronic myelofibrosis (PubMed database, accessed 250,319). Thus, this translocation seems to be most common in T-ALL, but it is overall uncommon, and a large clinical study reported the occurrence of this translocation in only 6% of adult T-ALL [133]. The translocation is associated with an adverse prognosis in T-ALL patient receiving treatment based on corticosteroids plus conventional chemotherapy [133].

### 6.1. The Structure and Function of the Fusion Partner SET (SET Nuclear Proto-Oncogene)

The SET-encoded protein is regarded as a histone chaperone; it forms dimers and each of the subunits consists of (i) an N terminus dimerization domain, (ii) an “earmuff” domain that is engaged in its histone chaperone activity and has binding sites both for histones and double-stranded DNA, and (iii) a negatively charged and acidic C-terminal domain (Figure 4) [136,137]. In this chapter, we give an overview of important SET functions, but its nuclear chaperone function is described in a separate chapter (Section 6.3).

The SET protein inhibits nucleosomal acetylation, especially of histone H4, by histone acetylases [136]. This inhibition is possibly caused by the masking of histone lysines from being acetylated, and histone acetyltransferase-dependent transcription is thereby silenced. The protein is also involved in regulation of histone methylation [136]. The SET protein can in addition be a part of a molecular complex localized to the endoplasmic reticulum, and it can have an antiapoptotic effect [137].

SET may contribute to leukemogenesis [137]; this hypothesis is suggested by the observation that mutations in the SET binding protein are associated with transformation of MDS to AML at least in certain genetic contexts. Increased expression of HOXA genes is observed as a result of this leukemic transformation (see Section 4).

SET binding protein 1 (SETBP1) is mainly localized to the nucleus, it binds the SET oncoprotein and the resulting heterodimer interacts with the PP2A serine/threonine phosphatase; this triple complex results in PP2A inhibition and AKT activation [138,139,140,141]. SETBP1 overexpression promotes the self-renewal of murine progenitors through regulation of HOXA9 and HOXA10 [142]. Furthermore, SETBP1 mutations inhibit protein degradation, and SETBP1 overexpression seems to facilitate leukemic transformation in ASXL1 mutated MDS through inhibition of PP2A activity and thereby AKT activation, repressed TGFβ, increased HOXA gene expression and induction of a molecular stem cell signature [143]. The final functional effect was then an antiapoptotic effect together with a differentiation block and finally increased proliferation. Thus, these effects show similarities with the effects of the SET-NUP214 fusion protein, suggesting that such effects are (at least partly) mediated by the SET component of the fusion protein. These molecular interactions may also support the hypothesis that the SET-NUP214 translocation has an adverse prognostic impact in AML; this is discussed in more detail in the last chapter of Section 6.5.

SET is a regulator of the p53 tumor suppressor [144]; its p53 binding depends on the acetylation status of the C-terminus p53 domain. SET inhibits the p53 transcriptional activity in unstressed cells, but this SET-mediated repression is abolished by stress-induced acetylation of the p53 C-terminus domain. Thus, loss of this SET interaction activates p53.

The transcription factor FOXO1 is acetylated by histone acetyl transferase; SET inhibits the p300-mediated FOXO1 acetylation and thereby allows increased DNA affinity with increased transcriptional activity [145]. This includes increased transcription of downstream gene p21 (a cyclin-dependent kinase inhibitor) in response to oxidative stress. On the other hand, stress-induced p53 activation also leads to increased p21 expression and thereby G1 cell cycle arrest or a chronic state of senescence or apoptosis [146]. These two examples illustrate that SET modulates p21 expression through different indirect mechanisms including both FOXO1 and p53 dependent mechanisms.

p53 is important for chemosensitivity in human AML, and p53 mutations are associated with adverse prognosis [125]. Wild-type p53 is a regulator of p21, and high constitutive p21 expression also seems to be associated with chemoresistance in human AML even though p21 and p53 levels show no significant correlation in primary AML cells [147]. However, p53 should be regarded as only one out of several p21 regulators [147], and the functional effects of p21 are also dependent on its biological context [146,148,149]. It can act as an oncogenic protein depending on, for example, the p53 status, growth factor deprivation, subcellular localization and during monocytic differentiation [150], but it can also function as a tumor suppressor and act as a cell cycle inhibitor. Thus, SET/p21/p53 interactions regulate complex cellular functions that are difficult to predict in individual patients.

SET is involved in the regulation of miR-137 [136], a mediator that is also involved in the regulation of AML cell proliferation/differentiation [151,152].

### 6.2. The Histone Chaperone Function of SET: An Important Function That Is Shared with DEK

A histone chaperone associates with free histones, prevents improper interactions with DNA and facilitates accurate ATP independent deposition of histones onto DNA, i.e., has nucleosome assembly activity [153,154]. The nucleosome is the repetitive chromatinal structure that consists of the core histone octamer formed by two molecules each of the H2A, H2B, H3, and H4 histones; this octamer core structure is surrounded by 146 base pairs of DNA. Most histone chaperones show a preference for binding either H3/H4 or H2A/H2B histones [153,154]. Histone chaperones are also involved in the removal of histones from nucleosomes that is required during DNA replication, repair and transcription (for references, see [153]). The two other classes of proteins that alter the histone structure being histone-modifying enzymes and ATP-dependent chromatin-remodeling complexes. Both DEK (see Section 5.1.2) and SET [154] are classified as H3/H4 histone chaperone [154].

SET is a histone chaperone that exists in two isoforms that are identical except for a short N-terminal sequence (including amino acids the complete 1–37 sequence or only the 1–24 sequence, respectively). The 1–37 amino acid isoform can bind all four nucleosome histones, but it preferentially binds the H3/H4 histones [155]. The H3/H4 binding activity is localized to the aminoterminal 1–225 amino acids [153]. The earmuff domain included in this aminoterminal section is also responsible for binding of both core histones and double-stranded DNA and for dimerization [153]. SET binds specifically to unacetylated and hypoacetylated (but not hyperacetylated) histones [156,157], and it is a part of molecular complexes that inhibit acetyl transferases [158]. However, SET can also regulate gene expression through other mechanisms than binding to/interactions with histone acetylation; SET can bind to and thereby block the DNA binding capacity of certain transcription factors (e.g., KLF5, SP1) [159,160], interact with nuclear receptor-type transcription factor (e.g., hormonal and vitamin receptors) [161], and modulate transcription through its binding to adaptor protein involved in transcriptional regulation [162]. Thus, SET can modulate gene transcription both through interactions with histones, nuclear transcription factors, nuclear receptors and transcription-regulatory adaptor proteins.

The studies described above show that SET can bind to or interact with multiple molecular targets. It is also a multifunctional protein involved in regulation of transcription, translation, cell cycle progression, cell survival/apoptosis and possibly also malignant transformation; the different functional effects can then be mediated by targeting of different molecular mechanisms [154].

### 6.3. The Structure and Functions of the SET-NUP214 Fusion Protein

The fusion gene can be formed by submicroscopic del(9)(q34.11q34.13) or a balanced t(9;9)(q34;q34) [163,164]. The molecular structure of the fusion protein is indicated in Figure 4. Its structure has been characterized by transcript analyses [13,163,165,166,167]. One transcript has 552 bp and consists of *SET* exon 7 at the 5′end (1–270 among 277 amino acids) fused with the *NUP214* exon 17 at the 3′end (813–2090 among 2090 amino acids) [166,167], whereas an alternative fusion product with 393 bp was formed with a transcript of *SET* exon 7 fused to exon 18 of *NUP214*. This transcriptional variation due to alternative splicing has been detected both in primary acute leukemia cells and cell lines.

One important consequence of the SET-NUP214 fusion protein is the disturbance of nuclear protein export [13,97,164,168], and this seems to be mediated by the NUP214 part of the fusion protein [168]. It is possibly due to interactions between the fusion protein and the nuclear exporter, XPO1, because a major part of the interactome proteins express the classical XPO1 binding motif [169]. Experimental studies in a T-ALL cell line have shown that inhibition of SET-NUP214 expression has an antiproliferative effect and induces differentiation [164]. Studies of the fusion protein interactome have demonstrated that many of the interacting proteins can associate with the microtubule cytoskeleton, an observation suggesting that the fusion protein also affects microtubule organization [169].

The SET-NUP214 fusion protein disturbs the nuclear localization of proteins involved in nucleocytoplasmic transport (including the XPO1 exporter); this is reflected by the formation of nuclear bodies that disperse during XPO1 inhibition and at the same time the fusion protein then relocalizes throughout the nucleoplasm [170]. These bodies contain XPO1 as well as XPO-cargo proteins and certain nucleoporins [168]. A consequence of this modulated nuclear transport is disturbed NFκB signaling [90]: this effect will possibly alter the constitutive cytokine/chemokine release by primary AML cells and thereby influence the communication between AML cells and neighboring nonleukemic supporting cells in the bone marrow microenvironment [171].

The SET-NUP214 fusion protein interacts with the lysine methyltransferase 2A/KMT2A, an enzyme that functions as a transcriptional coactivator and thereby as a regulator of hematopoiesis through its histone methyltransferase activity [172,173]. KMT2A regulates the transcription of specific target genes, including many HOX genes. The SET-NUP214/KMT2A complex enhances the promoter activity of *HOX* genes, e.g., the *HOXA10* gene, and the authors conclude that this interaction is mediated by the SET portion of the fusion protein. Such mechanisms may also explain the effect of the SET protein on histone methylation [173] (see Section 6.1).

A recent study investigated the SET-NUP214 interactome using a proteomic approach [169]. As described previously, the SET-NUP214 fusion protein inhibits nuclear export and interacts with transcriptional regulators. In accordance with these observations, several of the interactome proteins are involved in transcription/translation/RNA processing, but a large group of proteins were also involved in mitochondrial respiratory electron transport.

### 6.4. Patient Characteristics and Biological Characteristics of Non-T Acute Leukemia Cells with SET-NUP214 Fusion

The SET-NUP214 fusion protein are recruited to HOXA and HOXB clusters by their binding to chromatin-bound XPO1 protein; the XPO1 molecule is required for this recruitment of the fusion protein to these clusters and the subsequent local recruitment of RNA polymerase and finally aberrant activation of HOX genes [32]. However, the XPO1-SET-NUP214 complex can also accumulate in other regions, including CDKN2C and BMI1. The colocalization of XPO1 and NUP214 could be inhibited by the XPO1 inhibitor Selinexor/KPT-330, but the sensitivity of HOX gene expression to this XPO1 inhibitor seems to differ between AML cell lines. Such binding of fusion proteins to chromatin-located XPO1 with subsequent HOX activation has also been observed for the NUP98-HOXA9 fusion protein and mutant NPM1 [32,33,174].

A previous study investigated the effect of the SET-NUP214 fusion protein in a transgenic murine model; SET-NUP214 expression was then under control of the GATA1 gene hematopoietic regulatory domain that is important in distinct hematopoietic cell subsets [175]. These mice later developed anemia, thrombocytopenia and splenomegaly; this was followed by the high level of a c-kit^+^Sca^−^Lin^−^ cell population in the bone marrow together with decreased levels of erythroid, megakaryocytic and B cell lineage cells. The authors concluded that the fusion protein blocked hematopoietic differentiation programs. Another murine cancer study also concluded that the fusion protein caused expansion of an early progenitor cell pool with partial depletion of lymphocytes but without development of leukemia [176]. The fusion protein also inhibited monocytic differentiation of an AML cell line, including vitamin D3-induced differentiation [176,177].

The experimental observations described above with decreased levels of lymphoid, erythroid and megakaryocytic progenitors in *SET-NUP214* models [175,176] are also consistent with the observations that the SET-NUP214 fusion proteins has not been detected in lymphoid, erythroid and megakaryocytic malignancies. A variation in the expression of differentiation markers and aberrant expression of lineage-associated differentiation markers seems to be common in *SET-NUP214* hematological malignancies [131,132,178]. SET-NUP214 AML patients and their leukemic cells then have the following characteristics [131,133,163,179,180,181]:*Age.* There is a dominance of males among published cases (16 out of 20 published cases) and most of these 20 patients were relatively young (median age 32 years, range 12–50 years).*Differentiation and stemness.* Many of these T-ALLs show expression of the myeloid markers CD13 and CD33 and several AMLs with aberrant cytoplasmic CD3 have also been described. The non-T cell variants of SET-NUP214 positive acute leukemias show a considerable diversity according to the WHO classification and include AML, acute undifferentiated leukemia, mixed phenotype acute leukemia and blast phase of CML. This is therefore a very heterogeneous AML group with regard to differentiation/phenotype; whereas, in contrast, the large majority of T-ALL with SET-NUP214 fusion share a common characteristic—TCRγδ expression [133]. Finally, the fusion protein modulate vitamin D receptor functions, including its function as a transcription factor [181], and vitamin D is also important for maintaining stemness both in normal and AML leukemogenesis [182].*Leukemization.* These patients show a wide variation with regard to leukemization; both decreased leukocyte levels as well as hyperleukocytosis have been described. A recent review described a median white blood cell count of 38.8 × 10^9^/L (range 0.56–283 × 10^9^/L).*Genetics.* The karyotype of SET-NPM214 patients seems to vary considerably; several cases with normal karyotype have been described but also combinations with other cytogenetic abnormalities (hyperdiploidy, various translocations) [131,163,179,180].

Taken together, these observations show that the non-T cell acute leukemias with the SET-NUP214 fusion are both very heterogeneous with regard to signs of lineage differentiation as well as genotype.

### 6.5. Does the Fusion Protein SET-NUP214 Have Any Prognostic Impact in Human AML?

A previous small clinical study including 11 SET-NUP214 positive adult T-ALL patients described that SET-NUP214 fusion is associated with both corticosteroid and chemotherapy resistance to early/initial antileukemic treatment, but despite this resistance, the majority of patients later achieve completed remission [133]. Nine of these patients were allotransplanted, and the overall survival of these 11 patients did not differ significantly from other T-ALL patients. Another study including 11 adult patients receiving myeloablative conditioning and allogeneic stem cell transplantation in first remission, showed a 3-year survival of 38.5% [183]. Taken together, these two small studies may suggest that SET-NUP214 fusion is associated with high-risk disease in T-ALL, but it should be emphasized that additional studies are definitely needed.

A recent experimental study showed that high expression of the SET protein was associated with resistance of a breast cancer cell line against the cytotoxic agent paclitaxel [184]. SET overexpression increased mRNA and protein levels of ABC transporters and the PI3K/Akt pathway. Most of the SET molecule is included in the SET-NUP214 fusion protein, and these observations suggest that SET (and possibly also SET-NUP214) associated chemoresistance can be seen in certain cancer cells and include resistance to various anticancer agents.

Even though certain chemotherapeutic agents (i.e., anthracyclines) are used in the initial treatment of both ALL and AML, it should also be emphasized that the prognostic impact of SET-NUP214 may depend on the biological context, and thereby differ between T-ALL and AML. SET-NUP214 AML even shows biological similarities with NPM1-positive AML (i.e., XPO1-dependendent activation of HOX genes [32]) that has a favorable prognosis [125]. Thus, it is not possible to judge whether SET-NUP214 has any prognostic impact either in adult T-ALL or AML?

The SETBP1 protein is a nuclear protein that binds to SET and thereby leads to the formation of the SETBP1-SET-PP2A (protein phosphatase 2A) triple complex that results in PP2A inhibition and thereby enhanced proliferation [138,139,140,141,142]. Both SETBP1 mutations [184,185] and high cellular SETBP1 levels [138,185] in the leukemic cells are associated with adverse prognosis in AML. SETBP1 promotes the self-renewal of progenitor cells through activation of HOXA9 and HOXA10 [142]. Furthermore, PP2A is a human tumor suppressor that inhibits cellular transformation through modulation of the activity of several signaling proteins that are important for malignant cell functions/characteristics [139,140,186,187]. PP2A inactivation seem to be a common event in AML, the mechanism being either hyperphosphorylation, deregulated expression of the SET inhibitor, overexpression of SETBP1 or downregulation of PP2A subunits [139,140,141]. Thus, SET is one of the important regulators of this leukemogenic mediator and taken together, these observations supports the hypothesis that SET-NUP214 AML is also associated with adverse prognosis due to interactions between the fusion protein and SETBP1/PP2A.

## 7. Episomal Amplification of Chromosome Arm 9q with Formation of the NUP214-ABL1 Fusion Protein

This is an uncommon genetic abnormality in AML where the fusion protein localizes to the nuclear pore complex [188].

### 7.1. The Structure and Function of ABL1

The *ABL1* gene encodes the nonreceptor ABL tyrosine kinase [188,189]. The encoded protein has two main parts; the N-terminal kinase part and the C-terminal location part that are separated by a proline-rich linker (PRL) (Figure 5) [189]. The kinase part has the three SH3 (Src homology 3 domain), SH2 (Src homology 2 domain) and kinase domains listed from the N-terminal end. The linker contains binding sites for the SH3 binding domains of several ABL substrates. The C-terminal localization part includes the three DNA binding HLB (HMG-like box) regions together with three Nuclear localization signals (NLS), and the most C-terminal actin binding domain that includes the Nuclear export signal (NES). It should be emphasized that actin is also found in the nucleus [190], i.e., both the DNA and the actin binding regions contribute to the nuclear localization. ABL shuttles between the nucleus and the cytoplasm [189,191]. The nuclear export signal is important for this shuttling and for the export through the nuclear pores [4]. The nuclear export of ABL can be initiated by exogenous signals, e.g., binding to extracellular matrix [191].

Exogenous ligands can bind to both the SH3, SH2, kinase and PRL parts of ABL in addition to the DNA and actin-binding domains [189,192,193,194,195]. More than 100 ligands have been identified [189,192]. The ABL kinase is activated by a wide range of exogenous mediators, including various cytokines (e.g., VEGF), as well as intrinsic signals like DNA damage and oxidative stress [196]. This activation leads to events that are referred to as restricted activation, i.e., each activation signal may activate/modulate only certain intracellular ABL proteins and the final event may thereby be phosphorylation of a limited number of sites. Different activation signals may thereby obtain different and even opposite effects, e.g., cytokine-initiated activation can cause survival/growth enhancement whereas DNA damage-initiated activation can lead to apoptotic cell death [189]. Thus, the molecular context and intracellular localization of activated ABL will influence the final functional event, and the localization may also be altered/modulated in response to activation (e.g., nuclear accumulation upon DNA damage) [189].

SH3-binding ABL substrates can initiate autoinhibition of this kinase, i.e., a process where SH3 binds an internal ABL motif and thereby locks ABL in an inactive state [197]. This autoinhibition thus represents an upstream regulatory mechanism of ABL. The autoinhibition can be further modulated by a SH2-dependent process where actin binding causes a further allosteric enforcement or stabilization of the autoinhibition [189,193,194]. Ligation of the kinase domain may have a similar effect [195].

ABL can phosphorylate a wide range of functionally diverse proteins including adaptors, other kinases, cytoskeletal proteins, transcription factors and chromatin modifiers [192]. ABL can thereby influence both RNA processing and histone modification/acetylation [198,199,200].

### 7.2. The Genetic Abnormalities in NUP-ABL1 Fusion AML

The NUP214-ABL1 fusion varies between patients [201]. Different exons of NUP214 (exons 23, 28, 29, 30, 31, 32 and 34) can be fused predominantly with exon 2 of ABL1 (26/30 reported cases, 87%); the four remaining fusions involved exon 3. The most frequent fusions were exon 31 of NUP214 to exon 2 of ABL1 (13/30, 43%) followed by exon 29 of NUP214 to exon 2 of ABL1 (5/30, 17%). This variation in the structure of the fusion protein, i.e., the various parts of the ABL and NUP214 molecules that are included in the fusion protein, probably led to a biological and clinical heterogeneity of ABL1-NUP214 AML.

These observations demonstrate that the *NUP214-ABL1* variant of AML shows a genetic heterogeneity that probably leads to a biological heterogeneity of these patients and possibly also a heterogeneity with regard to chemosensitivity/prognosis of these patients. The final effect of this translocation is thereby difficult to predict because the ABL part of the fusion protein varies, and patients will therefore probably vary with regards to ligand binding, degree of autoinhibition, compartmentalization, downstream events of activation and modulation of these functions by the NUP214 part of the fusion protein.

### 7.3. The Clinical Characteristics of NUP214-ABL1 Fusion AML

The translocation is most frequent in ALL, especially T-ALL and less frequent in B-ALL [201]. This last study reviewed 42 published cases of this translocation among 842 patients (5%) included in six studies, and an additional 18 patients that were described in case reports. The following observations were made [201]. First, the translocation was detected both in pediatric and adult patients, and it was more frequent among males. Second, despite the genetic heterogeneity, T-ALL patients with NUP214-ABL1 translocation usually present with high risk factors, i.e., high white blood cell count, mediastinal mass and extramedullary manifestations [202]. Finally, observations in a few patients suggest that imatinib, dasatinib or bosutinib can be useful in the treatment of these patients [188,189,202].

Very few cases of AML with NUP214-ABL1 fusion have been reported, but the available data suggest [203,204,205,206,207] the following:*Age.* This genetic abnormality has been detected both in pediatric and adult AML patients, but the abnormality has also been detected in patients with high-risk MDS.*Differentiation of the AML cells.* Cases with morphological signs of differentiation have been reported. The AML cells can express CD4, CD7, CD11b, CD11c, CD13, CD33, CD34, CD38, CD45, CD56, CD64, CD117, CD123, HLA-DR and myeloperoxidase.*Diagnosis.* This is a cryptic fusion; the diagnosis is made by molecular genetic analysis. The breakpoints vary between patients. Fusion genes with NUP214 breakpoints in exon 29 and 34 have been detected.*Genetics.* The abnormality can be detected in patients with normal karyotype as well as patient with additional/complex cytogenetic abnormalities. Mutations of *BRCA2*, *GATA1*, *EGLN1*, *TP53*, *IDH2* and *RAD50* as well as whole deletions of *DDX41*, *NPM1* and *RAD50* have been detected in combination with the *NUP214-ABL1* fusion.*Prognosis.* Long-term AML-free survival has been described in one patient.

To conclude, very few AML patients with this fusion have been reported, their biological characteristics seem to vary and achievement of complete hematological remission as well as long-term AML free survival is possible, although too few patients have been reported to allow estimation of AML relapse/resistance risk.

## 8. The der(5)t(5;9)(q35;q34) with the Sequestosome-1(SQSTM1)-NUP214 Fusion Protein

The Sequestome-1 (SQSTM1, also known as p62) protein is a multifunctional protein that binds ubiquitin [208,209,210,211,212]. The protein thereby seems to have a function in selective autophagy and stress signaling [208,209,210,211,212,213,214], and its molecular interactions with NF-κB signaling seems important for its effects on autophagy/stress responses/inflammation [215,216,217]. It is also a nucleocytoplasmic shuttling protein, and this function seems to depend on XPO1 [218].

The very uncommon SEQST1-NUP214 fusion protein includes only 374 amino acids. This protein consists of a short C-terminal section of NUP214, and this section contains FG motifs that mediate an interaction with XPO1 and seem important both for the transformational capacity of the fusion protein and its impairment of myeloid differentiation [218,219,220]. The SEQSTM1-derived part of the fusion molecule includes its Phox-BEM1 (PB1) domain that is involved in oligomerization, the zinc finger domain and the TB (TGFβ-binding protein-like) domain that possibly mediates protein–protein interactions [13,218]. The fusion protein seems to upregulate the expression of *HOXA* and *MEIS1* genes in hematopoietic progenitor cells. Finally, overexpression of this fusion protein in mice causes development of myeloid leukemia, and this development seems to depend on the FG motifs of the fusion protein.

The expression of the two fusion proteins SET-NUP214 and SQSTM1-NUP214 were compared in a recent cell line study [168]. Overexpressed SET-NUP214 localized to nuclear bodies that also recruited the nuclear exporter XPO1 as well as certain other nucleoporins, and the nuclear export of both protein and RNA was then disturbed. On the other hand, the SQSTM1-NUP14 fusion protein was mainly localized in the cytoplasm, but it also formed nuclear bodies and disturbed protein export but not RNA export. The NUP214 part of the fusion proteins seems to mediate the inhibition of the nuclear export, whereas the SET/SQSTM1 parts are important for the fusion protein localization and thereby the extent of export inhibition. Thus, despite a similar FG-mediated interaction with XPO1 in the nucleus, other molecular interactions vary, and thereby the biological functions of the two fusion proteins probably also differ.

Animal studies suggest that SQSTM1-NUP214 can cause cellular transformation and thereby myeloid leukemia with impaired myeloid differentiation and increased expression of *HOXA* and *MEIS1* genes; these particular gene expression alterations seem to be mediated through the binding to XPO1 [220].

This fusion protein seems to be very uncommon in human AML, and its possible prognostic impact in AML patients is not known [221,222].

## 9. NUP214-RAC1 Fusion

One previous case report described an AML patient with an *NUP214-RAC1* fusion/translocation in combination with a complex karyotype [223]. The exon 17 of *NUP214* was then fused to exon 2 of *RAC*. The reading frame of *NUP214* was not matched with *RAC1*, but the AML cells showed high expression of *RAC1*. It is not known whether this translocation contributes to leukemic transformation of chemosensitivity. However, the RAC1 GTPase seems to be important for AML cell proliferation, survival and migration, and pharmacological RAC1 inhibition has a proapoptotic effect in certain experimental AML models [224,225,226,227].

## 10. Discussion

In this review, we describe several rare AML variants that involve the nuclear pore protein NUP214. The five NUP214 fusion abnormalities represent only a subset of AML patients with genetic abnormalities affecting nuclear pore molecules, another important AML subset is patients with NUP98-oncofusions that generally have an adverse prognosis [19,228,229,230]. Furthermore, high AML cell expression of the nuclear pore molecule XPO1 is also associated with clinical chemosensitivity and adverse prognosis in human AML [231]. Taken together, these observations suggest an important role of the nuclear pore complex in AML leukemogenesis and clinical chemosensitivity.

Previous studies have identified several established and emerging hallmarks of cancer, including nonmutational epigenetic reprogramming, sustained proliferative signaling, resisting cell death, genomic instability, modulation of the vasculature and altered immunoregulation [232,233,234]. Both the DEK-NUP214 and the SET-NUP214 fusion proteins modulate several of these hallmarks (Table 3, Section 5 and Section 6), and these effects are therefore relevant for AML leukemogenesis. The possible effects of the other NUP214 translocations/fusion proteins on these hallmarks have not been investigated.

The DEK-NUP214 and SET-NUP214 fusion proteins show several similarities (Table 4, see Section 6 and Section 7 for details) both with regard to the NUP214 fusion partners, the NUP214 contribution to the fusion molecules and possibly also the chemosensitivity of immature leukemic cells (i.e., AML and ALL) with these translocations. The similarities suggest that the two fusion proteins may have a similar prognostic impact in human AML. There seems to be a general agreement that DEK-NUP214 has an adverse prognostic impact, but the SET-NUP214 is very rare, and its possible prognostic impact in AML is therefore not known. The same is true for the other *NUP214*-invoving abnormalities.

The three other NUP214 variants differ from the DEK-NUP214 and SET-NUP214 AML variants. Firstly, in contrast to these two NUP214 translocations the ABL1 fusion molecules are involved in intracellular signaling, and the ABL1-NUP214 AML variants are probably heterogeneous with regard to genetics/leukemogenesis/chemosensitivity (see Section 7). Secondly, even though autophagy can be involved in leukemogenesis and SQSTM1 is both a regulator of autophagy and myeloid differentiation in hematopoietic progenitors [211,235,236,237], it is not known how the *SQSTM1-NUP214* translocations participates in leukemogenesis. Finally, RAC as well as other GTPases can be involved in leukemogenesis and probably also chemosensitivity in human AML [224,225,226,227,238], but the biology of AML cells have with the *NUP214-RAC1* translocations has not been characterized in detail.

Several of the fusion proteins have in common a molecular interaction with XPO1 and nuclear export mechanisms. A possible strategy for targeted therapy is, therefore, the use of XPO1 inhibitors for these NUP214-translocated AML variants. This strategy is also supported by experimental evidence [10,96,168,170]. An alternative strategy may be STAT5 inhibition, especially for the NUP214-DEK variant [73,74,100,101,107,110]. Finally, DEK can be released extracellularly and function as a ligand for the chemokine receptor CXCR2 [90,91,92,93,94]. CXCR2 expression is regarded as an adverse prognostic marker in human AML [95,96,97,98]. The t(6;9) fusion protein includes most of the DEK protein (Figure 3), but it is not known whether the fusion protein shows a similar extracellular release and CXCR2 binding as the full-length DEK molecule. To the best of our knowledge, it is not whether or how extracellular DEK shows posttranscriptional modification, but if CXCR2 ligation by DEK or the DEK-NUP214 fusion protein has any role in AML leukemogenesis/chemosensitivity. This effect may be inhibited by targeting of CXCR2 [239,240].

## 11. Conclusions

NUP214-involving oncofusions represent a heterogeneous group of uncommon AML variants. The prognostic impact is known only for DEK-NUP214 that has an adverse impact in human AML.

## Figures and Tables

**Figure 1 cells-14-01461-f001:**
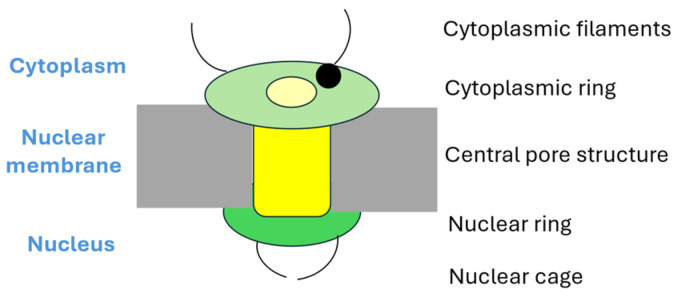
The schematic structure of the nuclear pore complex. This complex forms a pore through the nuclear membrane (grey color, see also the left part of the figure) and it consists of the central pore-forming complex (yellow) together with the cytoplasmic (light green) and the nuclear rings (dark green) that have attached cytoplasmic filaments and filaments forming the nuclear cage, respectively. The black circle indicates the position of NUP214 on the cytoplasmatic side of the pore and forming cytoplasmic filaments.

**Figure 2 cells-14-01461-f002:**
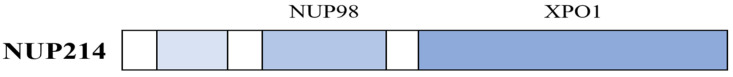
The main structure of NUP214 [14,15,16]. The molecule includes 2090 amino acids (see lower part). There are three main regions/domains; the β-propeller (left N-terminal, light blue), the coiled-coil (middle, blue) and the FG repeat domains (right, dark blue). The regions for binding of the two other nuclear pore complex molecules NUP89 and XPO1 are indicated at the top of the figure.

**Figure 3 cells-14-01461-f003:**
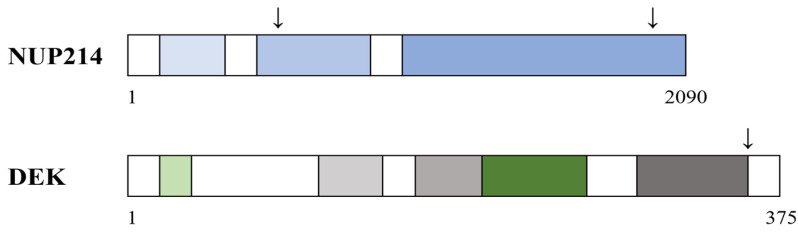
The main structure of NUP214 and DEK [13,47,53,54,55,56]. NUP214 includes 2090 amino acids and the three main regions/domains the β-propeller (left, light blue), coiled-coil (middle, blue) and the FG repeat domains (right, dark blue). DEK includes an aminoterminal (N) histone-binding region (light green) and a carboxyterminal region that mediates RNA binding activity and also includes a nuclear localization signal (dark green). The central region of DEK includes a pseudo-SAP domain (light grey) together with a neighboring DNA binding SAP domain (grey), and the carboxy(C)-terminal region binds DNA and facilitates DEK multimerization on chromatin (dark grey). The breaking points for the formation of the DEK-NUP214 fusion proteins are indicated by arrows. The fusion protein contains 42 out of the 44 FG repeats from NUP214. The N-terminal amino acids 1–349 of the DEK molecule are included, and the fusion molecule thereby contains 1626 amino acids.

**Figure 4 cells-14-01461-f004:**
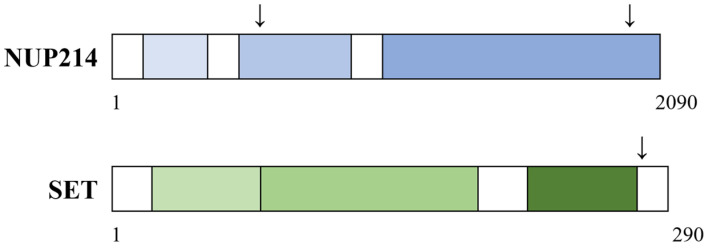
The main structure of NUP214 and SEK proteins [136,137]. NUP214 includes 2090 amino acids, and the three main regions/domains are the β-propeller (left, light blue), coiled-coil (middle, blue) and the FG repeat domains (right, dark blue). SET (290 amino acids) includes a dimerization (light green), an earmuff (green) and an acidic domain (dark green). The breaking points for formation of the SET-NUP214 fusion protein (1560 amino acids) are indicated by arrows. The fusion protein contains 42 out of the 44 FG repeats of NUP214. The C-terminal amino acids 1–349 of the SET molecule are included in the fusion protein that contains 1560 amino acids.

**Figure 5 cells-14-01461-f005:**
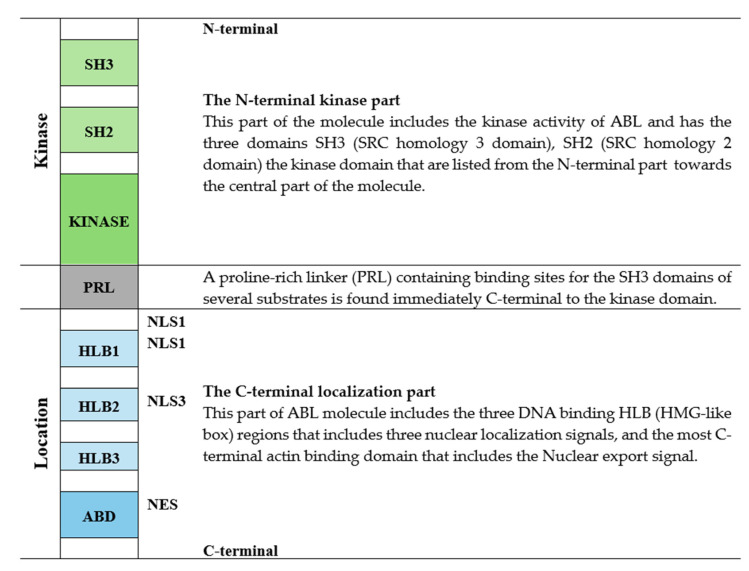
The molecular structure of the ABL1 protein [188,189]. The two main kinase and location regions (see left margin) are joined by the proline-rich linker (PRL) region. The N-terminal kinase region includes the three domains SH3, SH2 and kinase domains. The C-terminal part includes three DNA binding HLB domains that also include three nuclear localization signals (NLS) and the actin-binding domain (ABD) that includes the nuclear export signal (NES).

**Table 1 cells-14-01461-t001:** The molecular structure and phosphorylation of DEK molecules; an overview of the molecular regions and their phosphorylation. The table presents the various protein regions (AA, amino acids), their characteristics/functions (left column) and the importance of protein phosphorylation [47,53,54,55,56,57].

Molecular Region (Amino Acids) or Characteristic	The Name and Functional Characteristic of Each Molecular Region
AA 29–50, 248–246, 241–254, 299–310	The four **acidic regions** The (i) first region is located on the aminoterminal (N) side of the histone binding region, (ii) the two next regions are located between the SAP and the DNA binding carboxy(C)-terminal DNA binding domain where in the region that has RNA binding activity and a nuclear localization signal, and (iii) the last region is located on the aminoterminal side of the C-terminal DNA binding domain [55].
AA 52–60	The **histone binding region**; a region important for the histone chaperone function.
	A **pseudo-SAP domain** is localized close to the aminoterminal side of the SAP domain [47].
AA 149–187	This aminoterminal **SAP domain** has a DNA binding motif [47,53,54].
AA 270–350	**DNA binding and multimerization region** The C-terminal DNA binding domain is located in this part of the molecule that partly overlaps with the multimerization activity [53,54,55]. These activities are localized to the amino acid residues 270–350, and multimerization is in addition dependent on phosphorylation by CK2 [53].
Phosphorylation	DEK is a **phosphoprotein** [57] with several phosphorylation sites of which most are serine and threonine residues clustered in the carboxy-terminal region [53,54,55]. Multimerization as well as DNA binding by the C-terminal DNA binding domain are influenced by the phosphorylation status of DEK, and the multimerization is dependent on phosphorylation by CK2 [53,54,55]. The cellular levels of DEK are not altered but the DEK phosphorylation status varies during the cell cycle [55].

**Table 2 cells-14-01461-t002:** Cellular functions of the DEK proto-oncogene [54,55,56,57,58,59,60,61,62,63,64,65,66,67,68,69,70,71,72,73,74,75,76,77,78,79,80,81,82,83].

**DNA binding, posttranscriptional modulation with nuclear relocalization and DNA repair**
DEK is a DNA binding protein; its affinity to DNA depends on the phosphorylation status in its C-terminal part, but this phosphorylation does not influence its binding to chromatin [54]. Its DNA binding activity is also regulated by acetylation within the first 70 N-terminal amino acids that decreases the DNA affinity and leads to nuclear relocalization to RNA-processing structures [67]. DEK is important for several forms of DNA repair; the effects are partly mediated by RAD51 interactions [61,62].
**Chromatin modulation and histone interactions**
DEK shows hypophosphorylation and enhanced chromatin binding during apoptosis [68]. DEK can modulate histone H3 and to a lesser extent H4 acetylation through interactions with both histone deacetylases and histone acetyltransferases [58,59,60]. The acidic domain of DEK interacts with histones; DEK thereby inhibits histone transferase activity (e.g., p300 activity), induces histone hypoacetylation and reduces gene transcription [60].
**Transcription/splicing/translation**
DEK has an RNA binding domain that is possibly important for its role in RNA splicing [63,64,65,66], including altered splicing of genes involved in regulation of apoptosis and cell cycle progression [64]. It can decrease the expression of quiescence-associated proteins due to its modulation of histone acetylation/epigenetic regulation [58]. Studies in cancer cell lines have shown that DEK regulates the expression of several cancer-associated genes, including both oncogenes and tumor suppressors [64].
**Cellular effects: proliferation, adhesion, programmed cell death, quiescence, metabolism, signaling**
DEK modulates a wide range of genes involved in the regulation of cellular proliferation, adhesion/migration and DNA repair [62,69,70,71]. Animal studies suggest that DEK expression promote exit from quiescence [47,72]. DEK has an antiapoptotic effect; DEK depletion causes apoptosis and induces increased stabilization and transcriptional activity of p53 in cancer cell lines [73]. However, at least in certain models of FLT3-ITD AML, the DEK-mediated destabilization of p53 may be counteracted by STAT5-mediated reduction in p53 degradation [74]. The phosphorylation of DEK is altered as an early event during apoptosis [68]. DEK is involved in DNA repair, and the loss of DEK therefore induces genomic instability [62]. DEK preserves the self-renewing capacity of hematopoietic stem cells through increased quiescence and decreased mitochondrial metabolism; this effect partly depends on mTOR, but DEK also recruits the corepressor NCOR1 to reduce histone 3 acetylation and it thereby modulates expression of quiescence-associated genes [58]. Increased expression of the serine/threonine kinase IRAK1 in cancer cell lines [69]; this mediator is a downstream mediator of Toll-like receptors and the IL1 receptor that both are regulators of AML cell proliferation [75,76,77]. DEK has an antiapoptotic effect in cancer cell lines through increased IRAK1 expression [69]. DEK knockdown induces quiescence in normal hematopoietic cells and thereby renders these cells more resistant to irradiation [47,71].
**DEK in malignant cells and cancer-supporting bone marrow cells that are relevant for AML**
DEK mRNA and protein levels are low in primary AML cells compared with normal bone marrow cells [78]. DEK appears important for AML-supporting bone marrow stromal cells—it can stimulate cancer-associated angiogenesis and seems to stimulate M2-like macrophage polarization [70,79]. DEK expression in human breast cancer seems to induce a M2 macrophage phenotypes [70]. DEK increases VEGF expression in cancer cells through direct binding to the DEK-responsive element of the VEGF promoter and in addition through recruitment of HIF1α and the histone acetyltransferase p300 to the VEGF promoter [79]. DEK protein level, VEGF expression and microvessel density are correlated in human breast cancer [79].
**Extracellular release**
DEK is released extracellularly and functions as a chemokine or as a component of the neutrophil traps [80,81,82,83].

**Table 3 cells-14-01461-t003:** An overview of important biological functions of the fusion protein DEK-NUP214 [47,94,95,96,97,98,99,100,101,102,103,104,105,106].

**Transcriptional regulation and gene expression**
Transformation of human CD34^+^ cells by the fusion proteins causes increased HOXA and HOXB expression [102,103,104]. Primary AML cells with t(6;9) seem to have increased expression of EYA3, SESN1, PRDM2, and HIST2H4; these effects were observed together with the overexpression of *HOXA* and *HOXB* genes [103]. AML cells with t(6;9) AML show similarities in their gene regulation network with *NPM1-Ins* and *FLT3-ITD* positive AML, but they differs from these two variants with regard to the regulation of HOX genes [104]. The t(6;9) fusion proteins bind to and function as a XPO1-dependent transcriptional activator; this mechanism causes increased expression of several genes including *HOX* genes and *FOXC1* that are important in AML leukemogenesis [89]. The t(6;9) fusion protein also increases the expression of several other genes that seem to be involved in the process of AML leukemogenesis [96].
**Molecular interactions and localization (chromatin, DNA repair, transcription, RNA function)**
The t(6;9) fusion protein can bind to the nuclear pore protein XPO1; this is similar to the SET-NUP214 fusion protein (see Section 6) [94,95]. The fusion protein thereby functions as a XPO1-dependent transcriptional activator [96]. The DEK-NUP214 protein interacts with XPO1-mediated nuclear export of several proteins involved in leukemogenesis, including cyclin B1 and members of the NFκB signaling pathway [97]. The SET-NUP214 protein has a similar effect [97]. The fusion protein can also bind to several proteins involved in RNA processing and splicing, ribosome biogenesis, epigenetic regulation of gene expression, chromatine remodeling and DNA repair [95]. Some of these proteins have also been identified as DEK interacting proteins, whereas other DEK targeted proteins could not be identified as targets of the fusion protein (e.g., the kinase CK2 and histone H3) [47,95]. The fusion protein causes an altered localization of normal NUP214 away from the perinuclear region/nuclear pore towards an intranuclear microspeckled pattern [95]. The chromatin localization of full-length DEK was not altered by the fusion protein, except that binding to histone H3 was no longer detected [94,95]. The interactome of the DEK-NUP14 fusion protein includes mainly proteins that are involved in RNA metabolism and regulation of gene expression, but several genes involved in regulation of leukocyte activation, apoptosis and gene expression are also included in the interactome [95].
**Intracellular signaling**
Increased AML cell proliferation due to posttranscriptional upregulation of mTOR and increased mTOR1 activity [98,99]. The DEK-NUP214 fusion protein inhibits NFκB signaling [97]. The DEK-NUP214 fusion protein activates JAK2 and thereby upregulates STAT3 and STAT5 [100]. Studies in the human cell line FKH1 and cells transduced with the fusion protein showed that activation of STAT5, mTOR (i.e., mTOR1 activity with p70S6K phosphorylation); SRC family kinases were needed for continued growth and survival [95,99]. STAT5 activation can be further enhanced by genetic *FLT3* abnormalities [100,101].
**Cellular metabolism**
The increased mTORC1 activity also leads to a metabolic shift towards oxidative phosphorylation [102].
**Cellular functions: leukemogenesis and AML cell proliferation**
AML cells with t(6;9) show strong proliferative responses to G-CSF, GM-CSF and IL3 [98], but it should be emphasized that this study included very few patients. Murine studies suggest that the fusion protein causes cellular transformation and leukemic disease especially when long-term hematopoietic stem cells were transduced with the fusion protein; transduction of other immature hematopoietic stem cell subsets was less efficient [105]. Forced expression of the DEK-NUP214 fusion protein has a proliferation-promoting effect in AML cell lines; this effect seems to be caused by increased mTOR1 but not mTOR2 activity because it is associated with increased phosphorylation of the mTOR1 target p70S6K protein but not the mTOR2 target AKT at Ser 473 [99]. The fusion protein causes a global increase in protein synthesis, an effect that seems to be restricted to cells of the myeloid lineage and it correlates with the phosphorylation status of the translation initiating protein EIF4E [99,106].

**Table 4 cells-14-01461-t004:** Similarities between DEK-NUP214 and SET-NUP214 positive AML (see also Section 6 and Section 7).

The NUP214’s fusion partner	Most of the SET and DEK full-length molecules are included in the fusion molecules. Both fusion partners functions as H3/H4 histone chaperones and transcriptional regulators. Both partners function as p53 regulators.
The NUP part of the fusion gene/protein	The C-terminal XPO1 binding part of NUP214 is included in the fusion molecules Important effects on gene transcription are mediated by the binding to XPO1 by the NUP214 part of the fusion molecule. Their influence of HOX gene expression is probably mediated by NUP214. Both fusion molecules modulates nuclear export.
Prognostic impact	NUP-DEK is associated with adverse prognosis in human AML. NUP-SET is also associated with chemoresistance/adverse prognosis, but this is only documented in ALL.

## Data Availability

Not applicable.

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
