# Peer review of "NUP214 in Acute Myeloid Leukemia"

_cells, 2025, doi:10.3390/cells14181461_

Round 1
Reviewer 1 Report
Comments and Suggestions for Authors
This review discusses the NUP214 nucleoporin protein and its role in acute myeloid leukemias. It is an extensive, lengthy review that reads a bit like a chapter in a textbook. The content is very detailed and comprehensive, but unbalanced, with more content focused on DEK and DEK-NUP214 fusions than on the other NUP214 fusions. The organization and length could be vastly improved, with suggestions given below. Finally, the authors make several comments and brief descriptions of the similarities and differences between the various NUP214 fusion proteins, which would make an excellent, focused, and novel, review article and valuable resource to the field, but this is an underdeveloped aspect of the manuscript.
- There is heavy use of bullet points to list different definitions, facts, or concepts. This interrupts the readers’ train of thought for the content and is a main reason why it seems written like a book chapter. These need to be shortened, made into a graphical figure, and/or put into a narrative style of writing with the authors’ interpretations. Indeed, some of the information is already depicted in figure form (e.g., the first set of bullet points is partially summarized in figure 2), so it would help the reader to condense the information perhaps into the figure legends. By having these long bullet point lists, instead of integrating them into a narrative, the review becomes very long, which tires the reader and leads to them losing interest (a TL;DR “too long didn’t read” scenario)
- Figures 1 or 2 would benefit from having an additional panel that shows the localization of NUP214 as part of the nuclear pore complex.
- NUP214 is spelled incorrectly in lines 143 and 147. “Modulation” is misspelled in line 250. “Transplantation” is misspelled in line 455. “Clinical” is misspelled in line 511. “Lysines” is misspelled in line 581.
- Section 4, the discussion on Hox genes, is unrelated, or tangentially related, to a review focused on NUP214. It can be removed in the interest of shortening the manuscript. Likewise, the discussion on p21 and p53 in lines 609-624 and the mentions of NUP98 fusions.
- Figure 3, the image of the DEK protein, is missing the pseudo-SAP/SAP box domain and the recently identified histone binding region near the N-terminus.
- The tables, like the bullet points, are organized lists of facts that do not present information in a succinct, integrated way. Indeed, much of the content in the tables is repeated information from the bullet points listed above. For example, the second half of page 6 and Table 1 could be combined and presented as 1-2 paragraphs between sections 5.1 and 5.2. If the authors want to keep some tables, they are encouraged to display them as bullet-point lists with short descriptions (they don’t have to be full sentences).
- In lines 255-259, the author states that phosphorylated DEK is the secreted form that can function extracellularly. While it is true that phosphorylated DEK has weakened binding to DNA/chromatin, and CK2 inhibition leads to less secreted DEK, it has not yet been fully proven that secreted DEK is phosphorylated. The CK2 inhibitor may have other non-DEK consequences that impact DEK secretion. For example, the authors of that paper show the CK2 inhibitor leads to a general decrease in CD81 and exosome formation, which is a major mechanism for extracellular DEK. This is supported by the findings of Kappes et a,l which demonstrated that phosphorylated DEK may still be tethered to chromatin through dimerization with unphosphorylated DEK. Please clarify this section of the review article.
- The DEK section is missing a discussion of the H3.3 histone chaperone activity of DEK. Given that this is a common function between DEK and SET, it seems appropriate to mention it here.
- Overall, the authors write substantially more about DEK and non-fusion (full-length) DEK protein functions than the other NUP214 binding partners. Making the manuscript more balanced in content for each fusion partner would improve readability.
- Two sections discuss SETBP1 (627-638 and 759-772), and this reviewer thinks it would be easier for the reader to understand if they were combined into one paragraph.
- The authors make some insightful observations about the common molecular consequences and structural similarities between the various NUP214 fusion proteins, as well as their major differences. This represents a knowledge gap that has not been well defined elsewhere. The authors are encouraged to highlight these commonalities and differences more, perhaps with their own table (with multiple columns) and a section within the paper. Currently, it is only the focus of the Discussion section, which seems underdeveloped. A more in-depth compare/contrast analysis of the NUP214 fusions would be a novel perspective in this field and an important resource for NUP214-fusion hematopoietic malignancies, since most other reviews only discuss individual fusions or nucleoporins more broadly.
Author Response
This review discusses the NUP214 nucleoporin protein and its role in acute myeloid leukemias. It is an extensive, lengthy review that reads a bit like a chapter in a textbook. The content is very detailed and comprehensive, but unbalanced, with more content focused on DEK and DEK-NUP214 fusions than on the other NUP214 fusions. The organization and length could be vastly improved, with suggestions given below. Finally, the authors make several comments and brief descriptions of the similarities and differences between the various NUP214 fusion proteins, which would make an excellent, focused, and novel, review article and valuable resource to the field, but this is an underdeveloped aspect of the manuscript.
Comment: We agree that the sections describing DEK and the DEK-NUP214 fusion are longer than the other sections. We have therefore included additional data on the SET-NUP214 and ABL1-NUP214 variants of AML, but we would also emphasize that more studies are available than for the other variants, and this is a main reason why the DEK-NUP214 part is longer. However, we have extended the SET214 and NUP214-ABL1 parts of the article, whereas the descriptions of the two last translocations have not been changed because the abnormalities are very uncommon and few studies are available.
We now include a new small Table 3 where we emphasize the similarities between the DEK-NUP214 and SET-NUP214 abnormalities. We also include a new/rewritten chapter in the Discussion section where we compare the various NUP-214 variants (page 24-25). Eight new references have been included (references 235-240).
1.1. There is heavy use of bullet points to list different definitions, facts, or concepts. This interrupts the readers’ train of thought for the content and is a main reason why it seems written like a book chapter. These need to be shortened, made into a graphical figure, and/or put into a narrative style of writing with the authors’ interpretations. Indeed, some of the information is already depicted in figure form (e.g., the first set of bullet points is partially summarized in figure 2), so it would help the reader to condense the information perhaps into the figure legends. By having these long bullet point lists, instead of integrating them into a narrative, the review becomes very long, which tires the reader and leads to them losing interest (a TL;DR “too long didn’t read” scenario).
Response: We have reduced the use of bullet points. However, in some parts of the text where the overall length of the bullet point section is relatively short we have kept the points. The following alterations have been made:
• Sections 3.1 and 3.2. We have reduced the number of bullet points, and each point now has a title. For the first chapter below Figure 2 the bullet points have been left out, this part has been reorganized from bullet points to an ordinary text.
• Section 5.1.3. The points have been reorganized, the number of points is reduced, and a title is given for each point.
• Section 5.5. We do not use bullet points any more, the text is instead separated into different subsections 5.5.1-5.5.4 with informative headings for each subsection.
• Section 5.6. We have fewer bullet points, and we have inserted an informative short title for each point.
• Section 5.7. The number of points have been reduced; each point starts with an informative short title.
• Section 5.8. We have fewer bullet points; an informative short title is added for each point.
• Section 6.4. The number of points have been reduced; each point has a title.
We hope our solutions can be accepted.
1.2. Figures 1 or 2 would benefit from having an additional panel that shows the localization of NUP214 as part of the nuclear pore complex.
Response: We now indicate in Figure 1 the position of NUP214 in the nuclear pore complex. The nuclear pore is a complex molecular structure that includes more than 30 different molecules. In our opinion a detailed description of this complex structure is outside the scope of this article, but as stated in the text this can be found in several recent reviews.
For Figure 2 we have added location of the regions for binding of XPO1 and NUP98. The structure of NUP214 is described in detail in the text immediately below the figure.
We hope these solutions can be accepted.
1.3. NUP214 is spelled incorrectly in lines 143 and 147. “Modulation” is misspelled in line 250. “Transplantation” is misspelled in line 455. “Clinical” is misspelled in line 511. “Lysines” is misspelled in line 581.
Response: Both NUP214 have been corrected,, modulate/modulation has also been corrected (was probably line 594 in the original version), clinical is corrected, Lysines is corrected
1.4. Section 4, the discussion on Hox genes, is unrelated, or tangentially related, to a review focused on NUP214. It can be removed in the interest of shortening the manuscript. Likewise, the discussion on p21 and p53 in lines 609-624 and the mentions of NUP98 fusions.
Response: We think that the HOX section should be included because it describes a biological characteristic that is common to DEK-NUP214 and DEK-NUP214 AML as pointed out by the reviewer in comment 1.11. This is now emphasized in a new short comment at the end of Section 4, and in addition we have done a general shortening of the original parts of this section from 503 words in the original version to 400 words in our Revised Version.
In our opinion p21 deserves to be mentioned because p53 is important in SET-NUP214 AML. However, the p21 discussing chapter in Section 6.1 has been shortened from 214 words in the original version to 137 words in the Revised Version.
In our opinion NUP98 deserves to be mentioned because it is an important binding partner for NUP214 in the nuclear pore complex (see Section 3.2), and NUP98 fusion proteins also bind to XPO1 and thereby share a molecular mechanism with SET-NUP214 and DEK-NUP214 AML (Section 6.3). For this reason we think that our relatively short NUP98 comments throughout the review deserves to be included. We suppose that the reviewer especially refers to the summarizing bullet points before the conclusion of Section 3.2 (page 4); at the beginning of this chapter we have therefore added a new comment that clearly states that NUP98 anchors NUP214 to the nuclear pore complex.
We hope our solutions can be accepted.
1.5. Figure 3, the image of the DEK protein, is missing the pseudo-SAP/SAP box domain and the recently identified histone binding region near the N-terminus.
Response: We have corrected the figure as suggested by the reviewer. We apologize for the original incomplete figure. We have in addition added a new Table 1 where the molecular structure of DEK has been described in more detail, including the histone domain. New references have also been added together with the more detailed/complete descriptions of DEK (references 55-66).
1.6. The tables, like the bullet points, are organized lists of facts that do not present information in a succinct, integrated way. Indeed, much of the content in the tables is repeated information from the bullet points listed above. For example, the second half of page 6 and Table 1 could be combined and presented as 1-2 paragraphs between sections 5.1 and 5.2. If the authors want to keep some tables, they are encouraged to display them as bullet-point lists with short descriptions (they don’t have to be full sentences).
Response: The journal has a strict style with regard to the design of tables, and as we understand the guidelines bullet points are not allowed in the tables. The text should also be centered and not start at the left edge of the table, and in our opinion this would make bullet points to look a bit strange. Furthermore, the second reviewer made the following specific comments about the tables:
• “The Tables are well-structured and provide a good overview.”
• “On the other hand, the tables do provide an excellent overview.”
For these reasons we have not changed the tables. Our intention with the two large summarizing tables was to help the reader to have an easily available overview and a classification of the multiple and various complex cellular effects of DEK and DEK-NUP214. For these reasons we hope that the reviewer can accept our solution to keep the tables.
We would suggest that the Editors make the final decision on how to handle this comment.
1.7. In lines 255-259, the author states that phosphorylated DEK is the secreted form that can function extracellularly. While it is true that phosphorylated DEK has weakened binding to DNA/chromatin, and CK2 inhibition leads to less secreted DEK, it has not yet been fully proven that secreted DEK is phosphorylated. The CK2 inhibitor may have other non-DEK consequences that impact DEK secretion. For example, the authors of that paper show the CK2 inhibitor leads to a general decrease in CD81 and exosome formation, which is a major mechanism for extracellular DEK. This is supported by the findings of Kappes et al which demonstrated that phosphorylated DEK may still be tethered to chromatin through dimerization with unphosphorylated DEK. Please clarify this section of the review article.
Response: We agree with this comment. It is now clearly stated in Section 5.2 that it is not known whether or how secreted DEK shows posttranscriptional modifications (page 7).
This is further commented in the last chapter of the Discussion section (page) where we also comment on the possibility of CXCR2 targeting if such extracellular release is important for leukemogenesis and/or chemosensitivity in human AML (page 25).
1.8. The DEK section is missing a discussion of the H3.3 histone chaperone activity of DEK. Given that this is a common function between DEK and SET, it seems appropriate to mention it here.
Response: We have added new sections that describe the histone chaperone function of both DEK (Section 5.1.2) and SET (Section 6.3). New references have also been added (References 55-66 and references 154-162, respectively).
1.9. Overall, the authors write substantially more about DEK and non-fusion (full-length) DEK protein functions than the other NUP214 binding partners. Making the manuscript more balanced in content for each fusion partner would improve readability.
Response: We agree that there is an imbalance between the different NUP fusion partners. We have therefore added new sections that describe (i) the function of DET as a nuclear chaperone (Section 5.1.2), the function of SET as a nuclear chaperon (Section 6.3) and (iii) the structure and function of the ABL kinase (Section 7.1). We have not added new information about the two last fusion partners SQSTM1 and RAC1 because these abnormalities are very uncommon and for this reason they only deserve shorter presentations mainly to make the review complete.
One should also emphasize that the DEK-NUP214 variant of AML has been more extensively investigated, and for this reason the section about this variant is more extensive than the other ones.
We hope our solutions can be accepted.
1.10. Two sections discuss SETBP1 (627-638 and 759-772), and this reviewer thinks it would be easier for the reader to understand if they were combined into one paragraph.
Response: SETBP1 was actually commented on three places in the original version. We have now reduced this to two places. The text has not been altered, and these chapters are marked with grey in the Revised Version. We hope it can be accepted to comment on this in two separate chapters/Sections: one as a description of the SETBP1 biology in the Section describing the general function of SET, and the other part as a discussion of the possible prognostic impact of SETBP1/SET in AML. In each of these two parts we now refer to the other part.
We hope this solution can be accepted.
1.11. The authors make some insightful observations about the common molecular consequences and structural similarities between the various NUP214 fusion proteins, as well as their major differences. This represents a knowledge gap that has not been well defined elsewhere. The authors are encouraged to highlight these commonalities and differences more, perhaps with their own table (with multiple columns) and a section within the paper. Currently, it is only the focus of the Discussion section, which seems underdeveloped. A more in-depth compare/contrast analysis of the NUP214 fusions would be a novel perspective in this field and an important resource for NUP214-fusion hematopoietic malignancies, since most other reviews only discuss individual fusions or nucleoporins more broadly.
Response: Please see our response to the first comment 1.1 on page 1 of this letter. We have included a new Table 4 and a new/rewritten chapter in the Discussion section where we compare the DEK-NUP214 and SET-NUP214 variants of AML. This part of the Discussion section has been rewritten (see pages 24-25).
Reviewer 2 Report
Comments and Suggestions for Authors
Excellent review manuscript on a topic that has not been covered extensively. The Tables are well-structured and provide a good overview.
Although simplicity in figures is good, there appears to be some room for improvement.
A couple of concept images would be desirable (for the main interaction types and the main effects postulated), only if it is possible. On the other hand, the tables do provide an excellent overview.
The literature is mostly well-covered, with very few exceptions.
In summary, this is an excellent manuscript, with some room for improvement. Especially for the figures.
Author Response
2.1 Excellent review manuscript on a topic that has not been covered extensively. The Tables are well-structured and provide a good overview.
Response: We are very grateful for this general comment.
2.2 Although simplicity in figures is good, there appears to be some room for improvement. A couple of concept images would be desirable (for the main interaction types and the main effects postulated), only if it is possible. On the other hand, the tables do provide an excellent overview.
Response: Due to the statements about the tables in comment 2.1. and in this comment, we have not changed the tables even though this was suggested by reviewer 1. With regard to the figures, (i) we now indicate in Figure 1 the location of NUP214 in the nuclear pore complex, and (i) we now show the location of the binding of the two other nuclear pore molecules NUP98 and XPO1 (Figure 2) that are discussed more in detail in later parts of the review.
We have added a new Figure 5 that shows the structure of the ABL kinase.
2.3 The literature is mostly well-covered, with very few exceptions. In summary, this is an excellent manuscript, with some room for improvement. Especially for the figures.
Response: We are grateful for this general comment. We have now added new sections that describe more in detail (i) the structure and chromatin chaperone function of DEK (Section 5.1.2), (ii) the chromatin chaperon function of SET (Section 6.3), and (iii) structure of the AML kinase (Figure 5, Section 7.1). New references are added to these parts of the article.
New references have been added to describe DEK more in detail (references 55-66), the histone function of SET (references154-162), similarities of DEK-NUP214 and SET-NUP214 but not the other NUP214 translocations (references 235-242), and possible therapeutic strategies in these variants of AML (references 243 and 244).
Round 2
Reviewer 1 Report
Comments and Suggestions for Authors
The authors have sufficiently addressed the reviewers concerns.